



# Failure mode of rainfall-induced landslide of granite residual soil, southeastern Guangxi province, China

**Shanbai Wu**[1, 2, 3, 4]**, Ruihua Zhao**[1, 2, 3]**, Liping Liao**[1, 2, 3*]**, Yunchuan Yang**[1, 2, 3*]**, Yao Wei**[1, 2, 3]**, Wenzhi Wei**[1, 2, 3]

[1]College of Civil Engineering and Architecture, Guangxi University, Nanning 530004, China;

[2]Guangxi Key Laboratory of Disaster Prevention and Engineering Safety, Guangxi University, Nanning 530004, China;

[3]Key Laboratory of Disaster Prevention and Structural Safety of Ministry of Education, Guangxi University, Nanning 530004, China;

[4]Faculty of Engineering, China University of Geosciences, Wuhan 430074, China

**Correspondence:** Liping Liao (01llp@163.com), Yunchuan Yang (yyunchuan@163.com)

**Abstract.** Granite residual soil landslides are widely distributed in southeastern Guangxi province, China. They are posing a huge threat to local communities and hindering social and economic development. To understand the failure mode of the landslide can provide a scientific basis for early warning and prevention. In this study, it conducted artificial flume model tests to investigate the failure mode of granite residual soil landslide. The macroscopic phenomena of landslides in the flume were

summarized. The changes of soil moisture content along with pore water pressure were analyzed. And the differences and commonness in the initiation patterns of landslides were discussed. The results had four aspects. (1) There were significant similarities in the phenomenon of slope failures. In the beginning of the artificial rain, slopes were infiltrated, following by the slope toe soil softened and slipped. Another similar pattern was that continuous rainfall could cause soil crusts and runoff on the

slope surface. Short-term low-lying areas and interlocking ditches would appear due to surface runoff and rainwater erosion. (2) The increase of initial dry density enhanced the permeability resistance of rainwater to the residual soil, which led to a delay in the response time of water content and pore water pressure, and a decrease in pore water pressure. Moreover, the fluctuation characteristics of pore water pressure may be related to the type of soil shear deformation. (3) The starting time of a landslide was

delayed as the initial dry density and slope angle increased, but it was shortened due to the increase in rainfall intensity. Meanwhile, the initiation pattern changed from a sudden sliding type to a progressive failure type due to the increase of initial dry density. (4) The failure process of the granite residual soil landslide could be classified into five stages: rainwater infiltration, soil sliding at the slope toe, the occurrence of surface runoff and erosion, the formation of a steep free face, and the upper soil sliding.

Above research results can provide valuable references for the prevention and warning of granite residual soil landslide in southeast Guangxi.

**Keywords:** Failure mode; Rainfall-induced; Granite residual soil landslide; Southeastern Guangxi province




## 1    Introduction

Rainfall-induced landslides are the most common geohazards in the tropical and subtropical areas covered by granite residual soil, such as Brazil (Lacerda, 2007; Coutinho et al., 2019), Singapore (Rezaur et al., 2003; Rahardjo et al., 2008; Harianto et al., 2012; Zhai et al., 2016; Zhang et al., 2019), Malaysia (Rahman et al., 2018), Korea (Kim et al., 2004; Pham et al., 2019), the southern (Jiao et al., 2005; Luo et al., 2021; Liu et al., 2021; Liu et al., 2020a; Liu et al., 2020b) and southeastern China (Xia et al., 2019; Yao et al., 2021; Shu et al., 2021; Zhao et al., 2021). Guangxi is located in southeastern China, where granite is concentrated in the southeast, and landslides occur frequently (Liao et al., 2019). Hot and rainy climatic conditions have caused strong weathering of the surface granite, giving birth to tens of thousands of residual soil. This provides a superior environment for the formation of landslides. Therefore, the southeastern Guangxi has been threatened by granite residual soil landslides for a long time.  Granite residual soil is a regional special soil (Ministry of Construction of the People's Republic of China, 2002). One reason is that it has the dual mechanical properties of cohesive soil and sandy soil. The other is that it exhibits an abnormal combination of poor physical properties (such as high liquid limit and large void ratio) and high-strength properties in a natural state (Chen et al., 2011). However, granite residual soil is extremely sensitive to rainfall, and is easy to disintegrate and soften, which will induce large-scale landslides (Dahal et al., 2008; Liu et al., 2020a; Zhang and Tang, 2013). Although shallow landslides are the main type (Rahardjo et al., 2008; Kim et al., 2004), they still have the characteristics of high frequency (Kim et al., 2015), suddenness and mass occurrence.

The failure mode of residual soil landslide is an important basis for disaster prevention and mitigation and early warning and prediction of landslide (Rezaur et al., 2003). In this regard, many scholars have conducted in-depth studies on granite residual soil landslide and other residual soil landslide through statistical analysis, model tests and numerical simulations. They classified the type of granite residual soil (Wu, 2006b) and studied on the physical mechanical properties (Zhu and Anderson, 1998; Chen et al., 2011; Zhang and Tang, 2013; Chen and Gong, 2014; Xia et al., 2019), engineering characteristic (Wu, 2006a; Xu et al., 2017) and microstructure (Li et al., 2017; Wang et al., 2018). The formation condition (Zhan et al., 2012; Zuo et al., 2015) and instability mode (Zhao and Hu, 2005; Dahal et al., 2008; Hu et al., 2009; Xu and Jian, 2017) of granite residual soil landslides were revealed. They found and confirmed that the failure mode of residual soil slope is different from that of homogeneous soil and rock slope, it includes arc slip, plane slip and front shear slip, but plane slip is dominant (Zhan et al., 2010; Fu et al., 2018). Its failure surface is parallel to the original slope (Kim et al., 2004). They also pointed out rainfall is the most important external triggering factor due to two aspects (Coutinho et al., 2019). One is the deepening of the wetting peak induced by rainfall infiltration (Kim et al., 2004). Second, the increase in soil water content and pore water pressure can lead to a decrease in slope stability (Gasmo et al., 2000; Rezaur et al., 2003; Rahardjo et al., 2005; Lacerda, 2007; Rahardjo et al., 2008). Thus, in the process of landslide formation, the variation of physical property parameters such as moisture, matric suction or pore pressure play an important role in the residual soil landslide (Kassim et al., 2012; Igwe and Fukuoka, 2014; Pham et al., 2019; Zhai et al., 2016). Rainfall triggered mechanisms focus on completely weathered granite fill slope in Hong Kong, China. They are static liquefaction (Chen et al., 2004) and the transition from slide to flow due to localized transient pore water pressure (Take et al., 2004). However, static liquefaction is impossible





due to unsaturated condition. Instead, local transient pore water pressure can induce the initially slip, which further triggers the high-speed slide (Take et al., 2004). Another finding is that the initial dry density (Mukhlisin et al., 2008) and slope angle (Liu et al., 2020a; Liu et al., 2020b) can affect the water permeability and control the formation of landslides (Xu et al., 2018). Many scholars have carried out related studies on the relationship between dry density of other types of soil (such as sandy soil, volcanic residual soil, and gravel soil) and the initiation of landslides. They found through model tests that the initial density can determine the stress-strain characteristics of the soil, and it corresponds to the initiation mechanism of dilation and contraction (Dai et al., 1999a; Dai et al., 1999b; Mckenna et al., 2011). The macroscopic phenomena corresponding to these two mechanisms are: saturated loose slopes will suddenly liquefy and flow quickly, while saturated dense slopes will creep slowly (Iverson et al., 2000). It can be seen that there is a significant difference in the sliding motion rate of sand landslides (Iverson, 2005). Especially when the dry density is optimal, the moving speed and sliding distance of the landslide are both maximums (Wang and Sassa, 2001). This is mainly because the initial dry density affects the soil-water interaction and soil permeability (Ng and Pang, 2000; Lin et al., 2009; Jiang et al., 2017). For example, high-density steep slopes are much more resistant to rainwater penetration than low-density gentle slopes (Xu et al., 2018). A gentle slope can lead to better accumulation of rainwater, a faster increase in water content, but a slower rate of soil collapse (Liu et al., 2020a; Liu et al., 2020b). Other scholars have further confirmed the above results through numerical simulations. That is, the initial dry density has a decisive influence on the movement accumulation and evolution process of the landslide, and there are also significant differences in the slip rate (Liang et al., 2017).

The above researches have pointed out the direction for the follow-up work. However, most of the conclusions related to failure process focus on gravel soil (Chen et al., 2017), sandy soil (Moriwaki et al., 2004; Huang et al., 2008; Huang and Yuin, 2010), fill slope (Chen et al., 2004; Take et al., 2004), clay soil (Elkamhawy et al., 2018) and loess slope (Tu et al., 2009). Moreover, the degree of development of granite weathering crust is closely related to the climate, topography and environment (Qu et al., 2000), its residual soil has significant heterogeneity characteristics in terms of thickness, physical and mechanical property (Rahardjo et al., 2002; Harianto et al., 2012). These special characteristics lead to the complex initiation modes of landslides (Calcaterra and Parise, 2005; Mukhlisin and Taha, 2012; Liu et al., 2020a; Xia et al., 2019). At present, the failure mode of granite residual soil landslides in southeast Guangxi has not been studied, which has brought challenges to the prevention and early warning of landslides. Therefore, some scientific issues need to be solved. For example, what are the differences and similarities of the failure process of granite residual soil landslides? How do the physical parameters of the residual soil change? In this paper, it conducted artificial flume model tests to resolve the above issues. Firstly, the macroscopic phenomena of landslide are observed and summarized. Then, the variation characteristics of soil moisture content and pore water pressure are analyzed. Finally, the differences and commonness of rainfall-induced landslide failure mode are discussed. In addition, the groundwater level in the study area is relatively deep and can be ignored. Therefore, the landslide initiation of the granite residual soil does not depend on the fluctuation of the groundwater level.



## 2    Field site and method

### 2.1    Field site

Rong county is a typical high-prone area of rainfall-induced landslide of granite residual soil in southeast Guangxi (Liao et al., 2019). It lies between longitude 110°15′00″-110°53′00″ E and latitude 22°27′00″-23°07′00″ N (Figure 1). The county covers an area of 2257 km$^2$, with an average annual rainfall 1737.4 mm a$^{-1}$. The rainy period is from April to September, and the rainfall in this period
accounts for 78.6 % of the average annual rainfall. The area of magmatic rocks is 1260.09 km$^2$, accounting for 55.83 % of the total area of the county. The lithology is mainly granite with an area 1219.06 km$^2$.

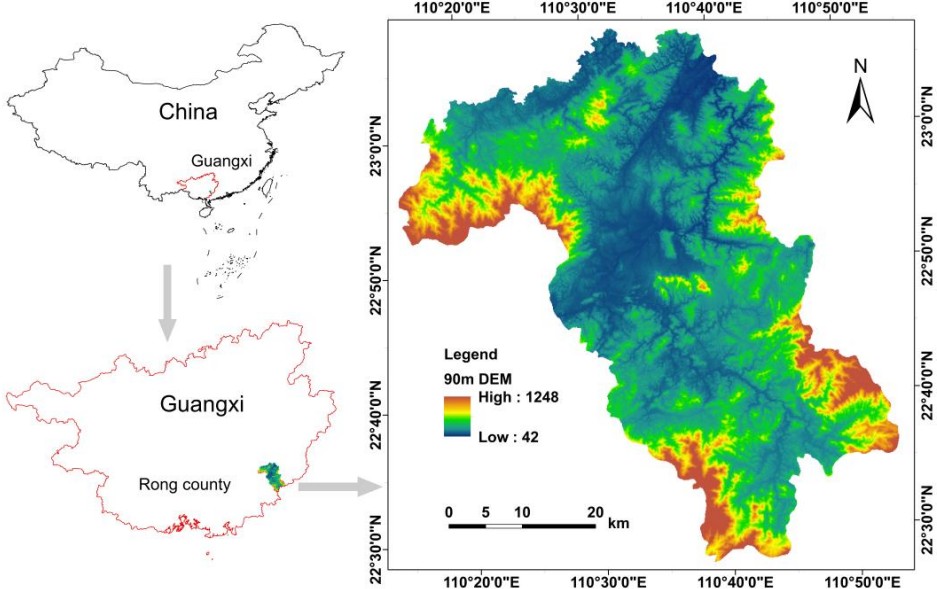

Figure 1 Study area.


### 2.2    Method

        Longtou village of Liuwang town is a landslide high-prone area in Rong county. Thus, the test soil is from Longtou village. Its specific gravity is 2.71, and the minimum and maximum dry density are 1.18 g cm$^{-3}$ and 1.72 g cm$^{-3}$ respectively. The particle data in Figure 2 is the average results of the
three groups of granite residual soil screening tests. Figure 2 shows that the cumulative content of gravel (diameter < 2 mm), and silt and clay (diameter ≤ 0.075 mm) are 87.52 % and 25.62 %. The angle of natural slope in study area is 30 to 45º. The dry density of surface soil is 1.20 to 1.40 g cm$^{-3}$, and the average mass moisture content is 6 %-10 % (Wen, 2015). Based on the above survey data, the test scheme in Table 1 sets only two initial dry densities: 1.20 g cm$^{-3}$ and 1.40 g cm$^{-3}$; the slope angle
is 40° and 45°; the initial mass moisture content is controlled in the range of 6 % to 10 %. Heavy rainfall is the main factor inducing landslide (Wei et al., 2017). Therefore, the test rainfall is set based on the rainfall data corresponding to landslides in 2010 (Wen, 2015). There are 2 to 3 periods of rainfall





in the experiment, and the duration of each period is 8 hours, with 15 hours between periods. Rain intensity is 60 mm h⁻¹, 90 mm h⁻¹.

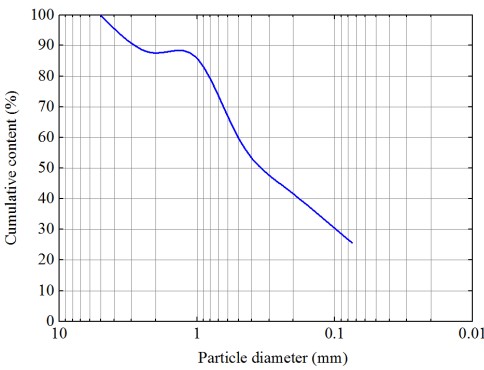


Figure 2 Particle gradation of granite residual soil.

Table 1 Sets of artificial flume model tests.

| Test number | Slope angle (°) | Initial dry density (g cm⁻³) | Rainfall intensity (mm h⁻¹) | Rainfall duration (h) |
|---|---|---|---|---|
| 1 | | 1.20 | 60 | 8, 8, 8 |
| 2 | 45 | 1.40 | 60 | 8, 8, 8 |
| 3 | | 1.20 | 90 | 8, 8 |
| 4 | | 1.40 | 90 | 8, 8 |
| 5 | 40 | 1.20 | 60 | 8, 8, 8 |
| 6 | | 1.20 | 90 | 8 |

The test devices are composed of rainfall control system, data testing system and flume model.
The rainfall control system includes the central control system, submersible pump, water tank, nozzle, bracket, and pipeline. The rainfall control system can set the number of large and small nozzles and their opening size. The height of the nozzle from the model is 2.3 m. The effective uniform rainfall area of this test is 6 m². As shown in Figure 3 and Figure 4, the testing system contains sensors and data collector. Because the unit of the acquisition time of the data collector is minutes, and the internal
storage capacity is limited, the collection frequency of volume moisture content and pore water pressure is 1 minute and 3 minutes respectively.

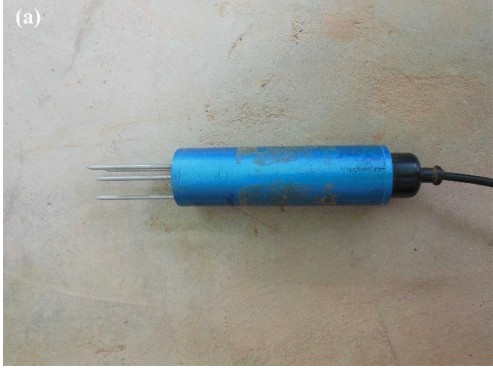

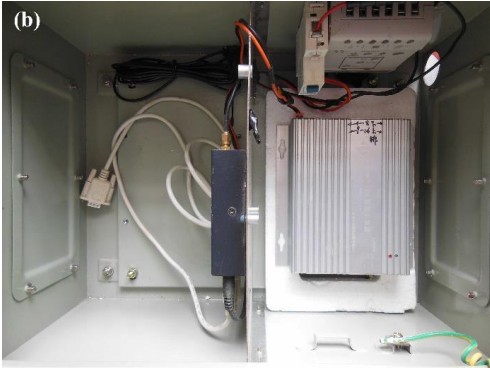

Figure 3 Volume moisture content testing system. (a) MP-406B soil moisture sensor. (b) M-16 data collector.



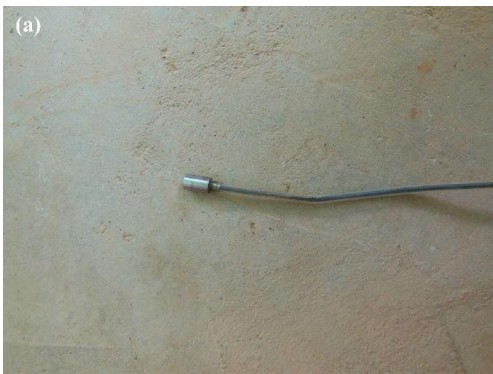
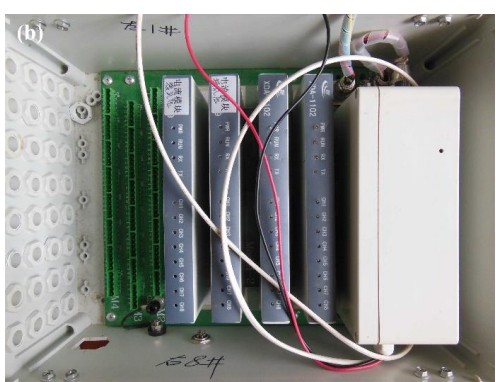

Figure 4 Pore water pressure testing system. (a) HC-25 micro pore water pressure gauge. (b) MCU data collection box.

The length, width, and height of the soil slope in the flume model are 1.5 m, 0.8 m, and 0.6 m (Figure 5). Before making the soil slope, a sufficient amount of air-dried soil samples are screened after being crushed. The required water is calculated based on the current and designed moisture

content. Then, they are sprayed evenly into the soil. When the soil is fully mixed, the soil is put into the container and kept for 24 hours. When the measured moisture content meets the requirements of the designed moisture content, the soil can be laid. The slope is divided into 6 layers. The thickness of each layer is 0.1 m. In order to achieve the preset initial dry density, each layer of soil sample is compacted with a wooden hammer. There are 12 monitoring points and 5 positions inside the slope

(Figure 5). Each monitoring point has a soil moisture sensor (the model is MP-406B) and a miniature pore water pressure sensor (the model is HC-25).

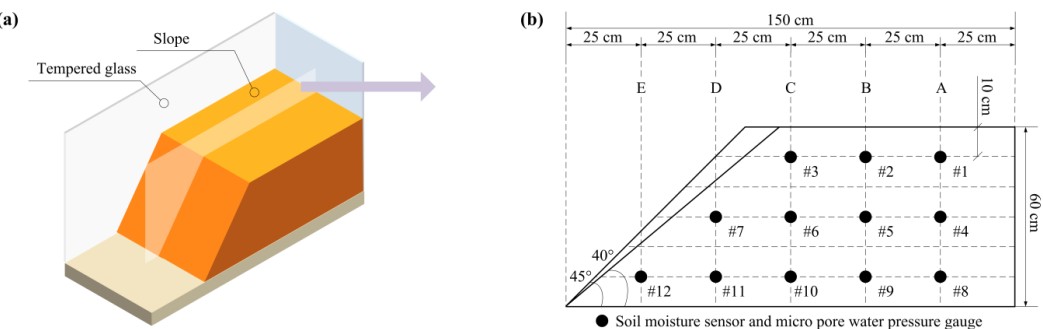

Figure 5 Test model and location of the instrument. (a) Schematic of the three-dimensional model. (b) Slope profile.

## 3    Results

### 175   3.1    Macroscopic phenomena of tests

(1) Test 1

During the first rainfall, when the rainfall lasts for 50 minutes, two small ditches are initially developed on the slope, and the soil at the foot of the slope slips, which in turn leads to the sliding of the soil in the slope. At this time, the instability area on the left side of the toe is fan-shaped, and the

length is three-quarters of the total length of the slope; the slope profile is concave. When the rainfall



lasts 421 minutes, a ditch developing on the slope shoulder is connected with the original instability area. During the second rainfall, the ditch is constantly washed by rainwater, and the fine particles are carried to the foot of the slope by rain. When the rain lasts for 559 minutes, the soil on the left shoulder slipped in blocks, causing the trailing edge to crack. Then, the surrounding soil slides and accumulates

to the foot of the slope. In the third rain, the rain lasted for 1324 minutes, and the continuous sliding resulted in the emergence of a steep surface; in addition, the left slope shoulder is washed by rain. Subsequently, the soil of the free surface slides. The soil sliding does not stop until the slope gradient becomes gentle (Figure 6a).

(2) Test 2

During the first rainfall, when the rainfall lasts for 67 minutes, the soil on the left side of the slope slips, and then drags the soil on the trailing edge to slide. When the rainfall lasts 431 minutes, the instability range has been extended to the slope shoulder, and the seventh sensor is exposed. Subsequently, the soil at the right toe slips slowly, causing the soil in the middle of the slope to slide. During the second rainfall, small cracks develop on the right slope. When the rainfall lasts for 524

minutes, the soil around the crack slips, and the sliding surface is arc-shaped. This process is repeated until the failure area extends to the slope shoulder, and the ditches form. The left slope is continuously washed by the third rain, and a ditch is formed and gradually connected with the right ditch (Figure 6b). At this time, the slope is in stable condition.

(3) Test 3

In the first rainfall process, when the rainfall lasts for 32 minutes, tensile cracks appear successively on the right and left slope toe, and the soil around the cracks slips. Subsequently, a steep free surface is formed. When the rainfall lasts for 39 minutes, the soil in the middle slope begins to slide. When the rainfall lasts for 215 minutes, the soil on the slope shoulder is pulled and then slips, which causes the sensor #3 to deviate from the embedded position. During the second rainfall, when

the rainfall lasts for 923 minutes, the soil in the middle of the slope slides gradually; it is followed by the sliding of the soil on the right shoulder. This sliding process is accompanied by the sinking of the slope (Figure 6c).

(4) Test 4

When the first rainfall lasts for 45 minutes, the left foot of the slope gradually slips. Muddy water

flows from the left foot. When the rainfall lasted for 78 minutes, the instability area on the left extended to the shoulders, while only a small amount of soil slipped on the right foot. During the second rainfall, the right slope is washed away by rain, which results in multiple low-lying areas. When the rainfall lasts for 496 minutes, although the soil of the right toe slides, the area is small and the slope is not completely destroyed (Figure 6d).

(5) Test 5

When the first rain lasts for 26 minutes, the soil on the right foot begins to slide, and the failure range gradually expands to the middle of the slope and the left foot. Slope erosion leads to the formation of low-lying areas. When the rainfall lasts for 208 minutes, the low-lying area becomes larger, and the soil at the foot has basically slipped. When the second rainfall lasts for 766 minutes, the

low-lying areas are connected, and a steep free surface is formed. When the rainfall lasts for 895 minutes, the soil at the foot continues to slide (Figure 6e). During the third rainfall, although the rain lasts 986 minutes, there is no obvious change in the slope.





Earth **Surface**
**Dynamics**
Discussions

(6) Test 6

When the first rain lasts for 5 minutes, cracks developed and expanded at the foot of the left slope,
which causes soil to slide down. When the rain lasts for 27 minutes, the sliding range has been extended
to the slope shoulder. Massive soil on free empty face slides from time to time. When the rain lasts for
96 minutes, the soil in the middle of the slope begins to slide, which causes the sensor #7 to be exposed
and the formation of a steep surface. When the rain lasts for 133 minutes, a large mass of soil on the
left shoulder begins to slip. When the rain lasts for 220 minutes, the soil at the right toe continues to
slide. When the rain lasts for 297 minutes, massive soil on the right side of the middle of slope slips
suddenly. At the end of the rain, the soil at the right shoulder remains stable (Figure 6f).

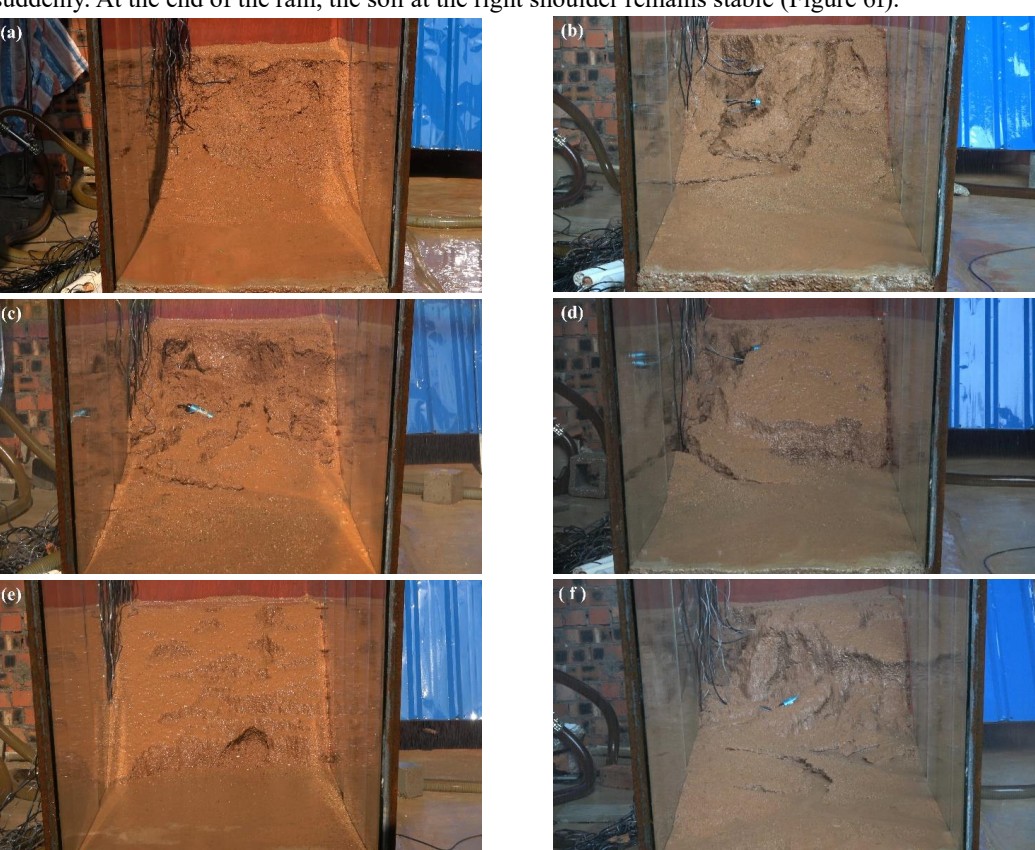

Figure 6 Slope after soil sliding in six sets of tests. (a) Test 1. (b) Test 2. (c) Test 3. (d) Test 4. (e) Test 5. (f) Test 6.

## 3.2    Volume moisture content

Because the change characteristics of the volume moisture content (VMC) at A and B are similar,
this paper only shows the VMC at four position B, C, D, E. Figure 7-9 shows that the VMC changes
at these four positions mainly include three stages: initial constant, significant rise, and stability.
However, there are still slight differences of each position. The VMC at the depth of 10 cm of the slope
top of the position B rise first, and the VMC at a depth of 30 cm and a depth of 50 cm increases
sequentially. The stable value of the VMC increases as the depth increases. Before the next stage of



rainfall begins, VMC is reduced due to evaporation; the second and third rainfall can restore VMC to
near the stable value of VMC during the first rainfall, with the smallest change in VMC at a depth of
50 cm. The changes in VMC at position C are similar to those at position B. However, during the first
rain in test 2, the stable values of the three depths are relatively close. In test 1, the VMC at the depth
of 30 cm at the position D begins to increase, and the VMC at the depth of 50 cm increases subsequently.
In test 2, the stable VMC at the depth of 30 cm at position D is similar to the value at the depth of 50
cm at position E.

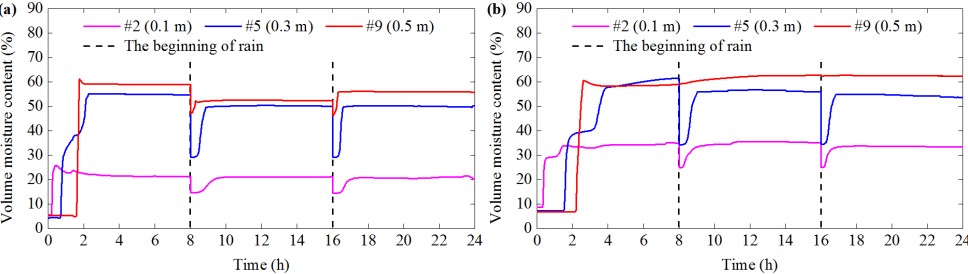

Figure 7 Volume moisture content at position B in (a) test 1 and (b) test 2.

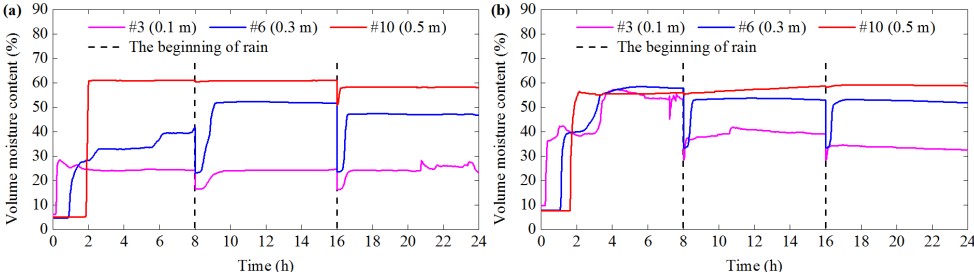

Figure 8 Volume moisture content at position C in (a) test 1 and (b) test 2.

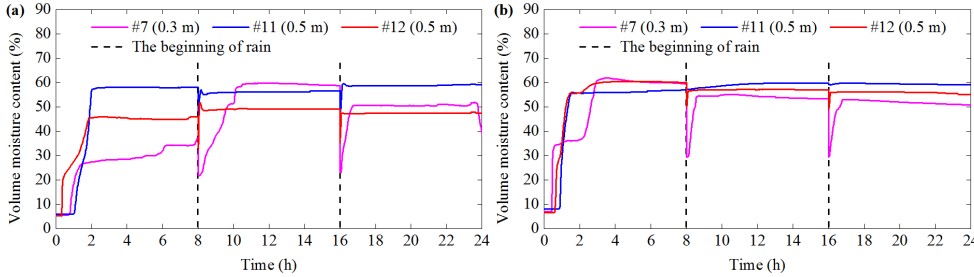

Figure 9 Volume moisture content at position D and E in (a) test 1 and (b) test 2.

The VMC of test 3 and test 4 is shown in Figure 10-12. At the beginning of the first rain, the
response time of VMC at position B is delayed as the depth increases, and the stable value of VMC
increases as the depth increases. The change of VMC at position C is similar to that at position B;
however, in test 3, the VMC at a depth of 10 cm remains stable, then decreases, and eventually becomes
zero. The reason is that the formation of landslide causes the third sensor to deviate from its original
position. The VMC with a depth of 30 cm at position D and the VMC with a depth of 50 cm at position
E increase simultaneously, and the stable values are not much different; in addition, the increase time
of VMC with a depth of 50 cm at position D is relatively late 15-30 minutes.



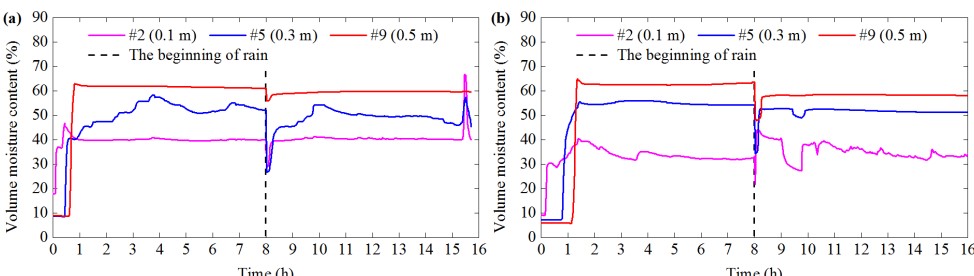

Figure 10 Volume moisture content at position B in (a) test 3 and (b) test 4.

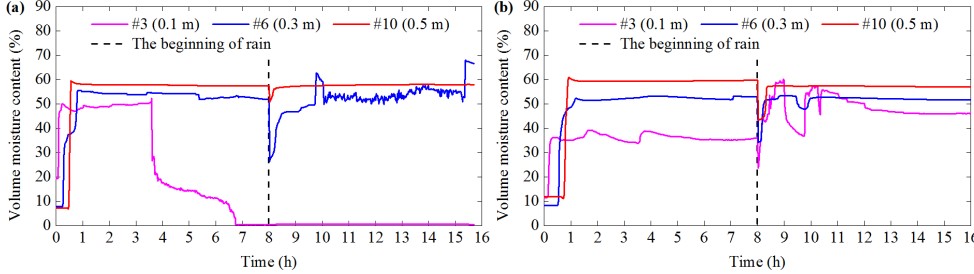

Figure 11 Volume moisture content at position C in (a) test 3 and (b) test 4.

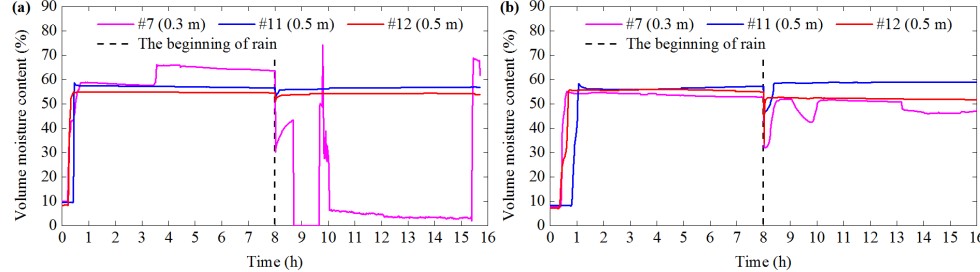

Figure 12 Volume moisture content at position D and E in (a) test 3 and (b) test 4.

The VMC of test 5 and test 6 is shown in Figure 13-15. In test 5, when the first rain lasts for 179 minutes, the VMC at a depth of 30 cm at position B begins to increase, while the VMC at a depth of 50 cm starts to increase significantly when the rain lasts for 105 minutes. The change in VMC at position C is similar to that at position B. The stable value of VMC at a depth of 50 cm at D and E is relatively close, but in Test 6, the stable value of VMC at a depth of 30 cm is the largest.

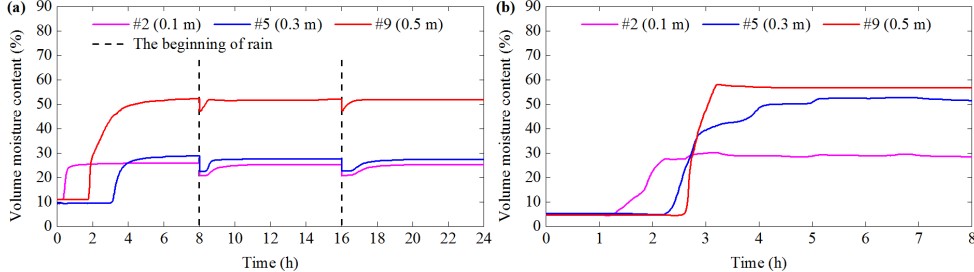

Figure 13 Volume moisture content at position B in (a) test 5 and (b) test 6.



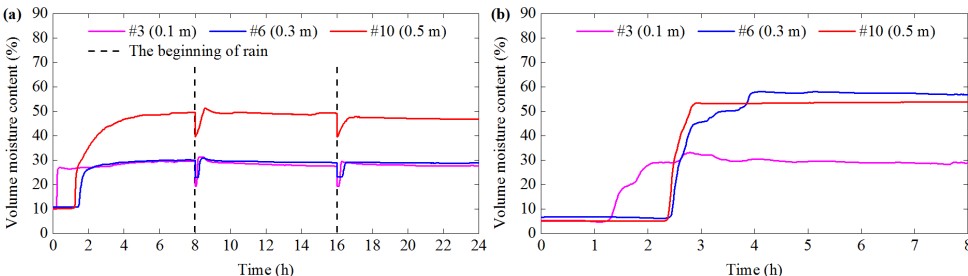

Figure 14 Volume moisture content at position C in (a) test 5 and (b) test 6.

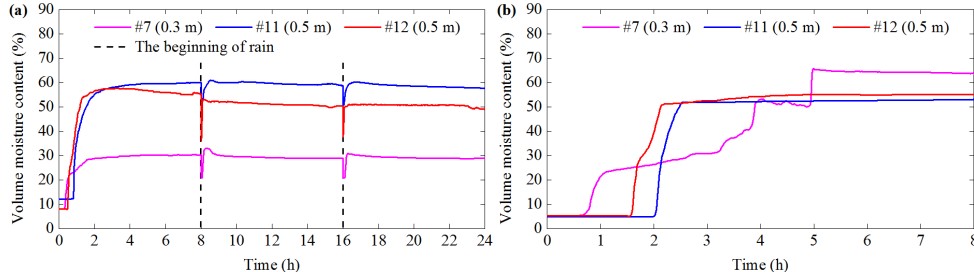

Figure 15 Volume moisture content at position D and E in (a) test 5 and (b) test 6.

Table 2 shows the response time and stable values of VMC at five positions during the first rainfall. For the position A, the VMC response time with the same depth is delayed with the initial dry density (IDD) increases and advanced with the increase of rainfall intensity, but the VMC response time of test 5 is earlier than that of test 6. In addition, the stable VMC increases with the increase of IDD, but decreases with the increase of rainfall intensity. The VMC response time and stable value of position B are similar to those of position A, but the stable VMC of test 3 at a depth of 10 cm is larger than that of test 4. For position C, the VMC response time and stable value of test 1 at a depth of 50 cm are greater than those of test 2. The VMC response time and stable value change at position E are similar to those of position A, but the VMC response time of position D in test 1 is longer than that in test 2.

Table 2 Response time and stable value of volume moisture content in six sets of tests under the first rainfall.

| Position number | Sensor number | Response time (min) | | | | | | Stable value (%) | | | | | |
| --- | --- | --- | --- | --- | --- | --- | --- | --- | --- | --- | --- | --- | --- |
| | | Test 1 | Test 2 | Test 3 | Test 4 | Test 5 | Test 6 | Test 1 | Test 2 | Test 3 | Test 4 | Test 5 | Test 6 |
| A | #1 | 13 | 24 | 3 | 7 | 21 | 135 | 17.2 | 29.5 | 33.0 | 22.6 | 23.2 | 24.6 |
| | #4 | 37 | 128 | 32 | 35 | 161 | 193 | 38.6 | 44.6 | 35.1 | 56.7 | 23.4 | 45.8 |
| | #8 | 74 | 171 | 52 | 84 | 152 | 172 | 57.3 | 60.8 | 63.5 | 64.9 | 47.9 | 58.9 |
| B | #2 | 13 | 18 | 3 | 9 | 21 | 75 | 21.3 | 33.6 | 39.7 | 32.4 | 25.9 | 29.2 |
| | #5 | 43 | 91 | 28 | 46 | 179 | 134 | 54.9 | 57.1 | 50.9 | 54.7 | 28.6 | 52.3 |
| | #9 | 96 | 129 | 40 | 68 | 105 | 155 | 58.9 | 60.4 | 61.6 | 62.8 | 51.5 | 56.7 |
| C | #3 | 13 | 16 | 3 | 9 | 14 | 76 | 24.2 | 52.8 | 49.5 | 35.7 | 29.2 | 29.6 |
| | #6 | 52 | 64 | 16 | 30 | 84 | 139 | 42.6 | 58.3 | 54.1 | 51.5 | 30.1 | 57.3 |
| | #10 | 111 | 103 | 29 | 40 | 73 | 142 | 60.8 | 56.5 | 57.9 | 59.3 | 48.4 | 53.6 |
| D | #7 | 46 | 25 | 15 | 24 | 20 | 40 | 30.4 | 60.6 | 58.6 | 66.1 | 30.1 | 64.3 |
| | #11 | 61 | 52 | 28 | 47 | 48 | 121 | 58.1 | 55.8 | 56.1 | 57.3 | 60.0 | 52.5 |
| E | #12 | 19 | 35 | 15 | 21 | 31 | 95 | 45.8 | 59.7 | 54.9 | 55.4 | 57.5 | 55.1 |





The results of the VMC show that under the same rainfall intensity and slope angle, the VMC response time and stable value for IDD of 1.20 g cm$^{-3}$ is less than that of 1.40 g cm$^{-3}$. This is because the increase of IDD will hinder the infiltration of rainwater and increase the water-holding capacity of the granite residual soil (Lee et al., 2005; Lu et al., 2018). In addition, when the IDD and slope angle are the same, the increase in rainfall intensity can shorten the response time of VMC and increase the stable value of VMC, which is due to the significant increase in rainwater infiltration. However, the response time of VMC in test 6 is mostly longer than that in test 5. It may be caused by premature accumulation of water at the foot of the slope due to heavy rain. The VMC stable value at a depth of 50 cm is much higher than that at a depth of 10 cm. It indicates that the soil at the slope toe is over-saturated. The reason is that the fine particles in the shallow soil move with the rain to the deep soil; this process will cause the infiltration path to be blocked or destroyed, which in turn leads to the formation of temporary stagnant water surfaces or barriers (Fang et al., 2012). In addition, the turbid water flowing from the foot of the slope indicates that the water is mixed with fine particles. This phenomenon once again confirms the fact that fine particles have migrated.

### 3.3    Pore water pressure

In order to discuss the relationship between landslide failure, soil moisture content and pore water pressure, this paper only shows the pore water pressure (PWP) at three positions (C, D, E) closely related to landslide. Figure 16-17 shows that PWP changes mainly include four stages: initial slow change, significant increase, dynamic fluctuation and stability. However, there are differences in PWP changes. In test 1, the PWP at a depth of 30 cm at position C fluctuates drastically in two periods: 3 hours, 30 minutes to 5 hours, and 5 hours, 50 minutes to 6 hours. In test 2, the PWP at a depth of 30 cm and 50 cm at position C remains stable after increasing; during this process, the PWP does not fluctuate sharply, and the change range is smaller than that in test 1. In test 1, the change of PWP with a depth of 30 cm at position D is relatively gentle; the PWP with a depth of 30 cm at position D and E turns from a negative value to a positive value in the first rain period. In test 2, the PWP with a depth of 30 cm at position D first increases and then decreases; the PWP with a depth of 50 cm at position D and E both gradually increases, and finally remains stable.

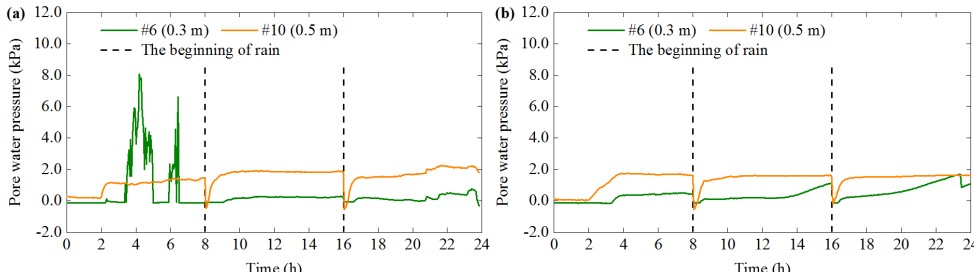

Figure 16 Pore water pressure at position C in (a) test 1 and (b) test 2.



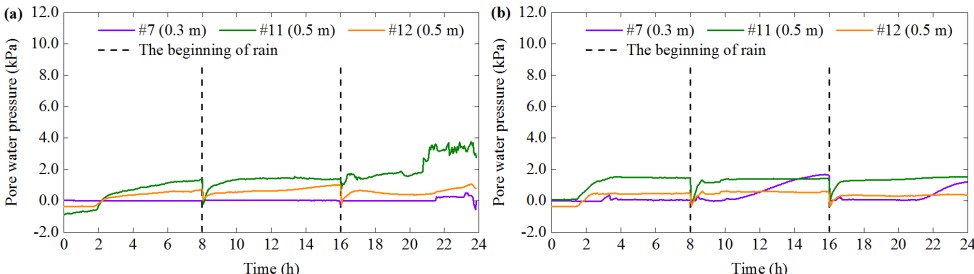

Figure 17 Pore water pressure at position D and E in (a) test 1 and (b) test 2.

The PWP of test 3 and test 4 is shown in Figure 18-19. In test 3 and test 4, the PWP change at a
depth of 30 cm and 50 cm at position C contains four stages: constant, increase, decrease and
fluctuation. However, the drop in PWP at a depth of 30 cm is greater than that at a depth of 50 cm. In
test 3, the PWP at the depth of 30 cm and 50 cm at position D shows an upward trend. Among them,
the PWP at a depth of 50 cm increases the most; the PWP at a depth of 50 cm at position E first
increases and then decreases. In test 4, the PWP at the depth of 50 cm at position E increases first, and
the PWP at the depth of 30 cm and 50 cm at position D increases sequentially, and the peak value of
pore pressure at the depth of 50 cm is the largest.

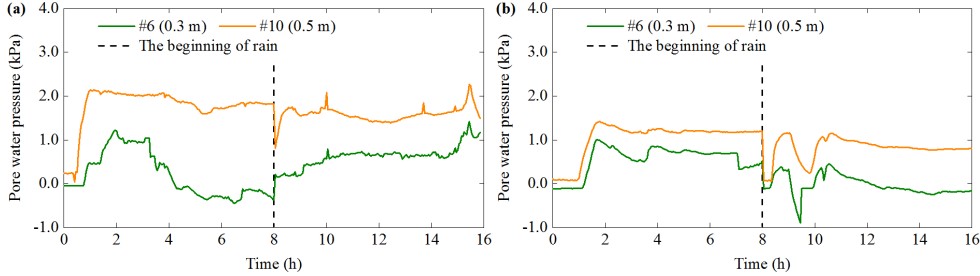

Figure 18 Pore water pressure at position C in (a) test 3 and (b) test 4.

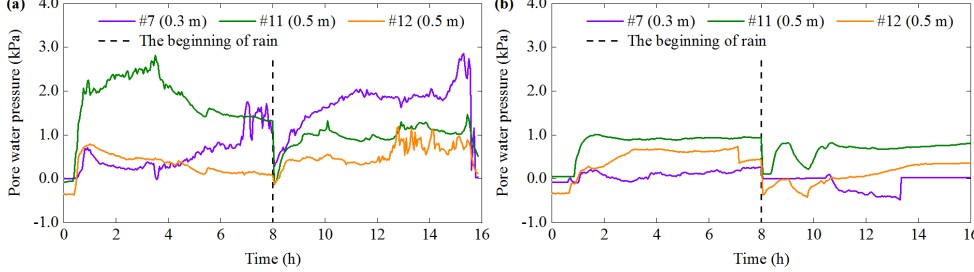

Figure 19 Pore water pressure at position D and E in (a) test 3 and (b) test 4.

The changes in PWP of test 5 and test 6 are shown in Figure 20-21. For the depth of 50 cm of test
5, the PWP changes gently in the first two rainfall stages, but it increases significantly in the third
rainfall. In test 6, the PWP at a depth of 30 cm and 50 cm gradually decreases after increases. For
positions D and E in test 5, the PWP at a depth of 30 cm does not change much, but the PWP at the
depth of 50 cm changes significantly. The PWP changes of test 6 are similar to those of position C.





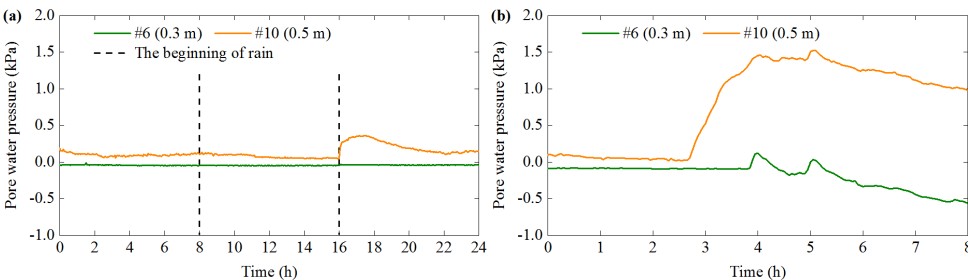

Figure 20 Pore water pressure at position C in (a) test 5 and (b) test 6.

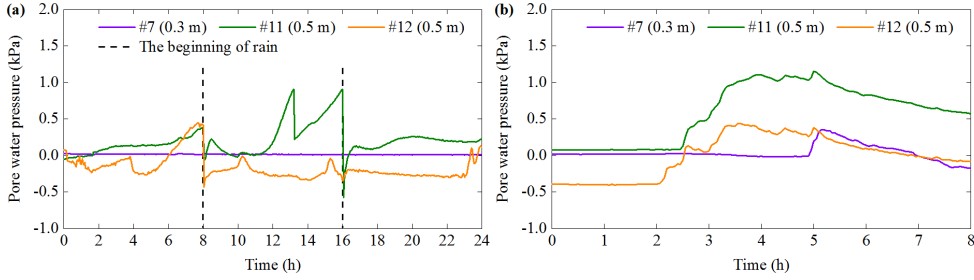

Figure 21 Pore water pressure at position D and E in (a) test 5 and (b) test 6.

Table 3 shows the response time and variation of PWP in the six tests during the first rainfall. The PWP at position A responds quickly at the depths of 10 cm and 30 cm; the response time of the PWP at the depth of 50 cm is delayed with the increase of IDD, but the PWP increases with the increase of

rainfall intensity. The changes of PWP response time and variation at position B are similar to those at position A, but the PWP variation in the depth of 30 cm in test 5 is greater than that in test 6. For the position C, the response time of PWP in test 5 is shorter than the time in test 6. The changes of PWP response time and variation at position D and E are similar to those at position C.

Table 3 Response time and variation of pore water pressure in six sets of tests under the first rainfall.

| Position number | Sensor number | Response time (min) | | | | | | Variation (kPa) | | | | | |
|---|---|---|---|---|---|---|---|---|---|---|---|---|---|
| | | Test 1 | Test 2 | Test 3 | Test 4 | Test 5 | Test 6 | Test 1 | Test 2 | Test 3 | Test 4 | Test 5 | Test 6 |
| A | #1 | 3 | 3 | 3 | 3 | 9 | 6 | 0.042 | 0.028 | 1.233 | 0.107 | 0.045 | 0.052 |
| | #4 | 6 | 6 | 6 | 9 | 18 | 9 | 2.451 | 0.781 | 0.554 | 2.537 | 0.569 | 0.614 |
| | #8 | 87 | 174 | 45 | 78 | 369 | 207 | 1.467 | 1.129 | 3.015 | 1.419 | 0.095 | 0.953 |
| B | #2 | 3 | 18 | 15 | 21 | 3 | 3 | 0.106 | 0.085 | 2.513 | 0.087 | 0.097 | 0.102 |
| | #5 | 21 | 78 | 36 | 72 | 15 | 9 | 0.494 | 0.210 | 0.847 | 0.796 | 0.089 | 0.038 |
| | #9 | 96 | 135 | 42 | 69 | 36 | 21 | 1.901 | 1.753 | 2.678 | 1.562 | 0.472 | 1.894 |
| C | #3 | — | — | — | — | — | — | — | — | — | — | — | — |
| | #6 | 24 | 192 | 24 | 63 | 3 | 228 | 8.141 | 0.656 | 1.639 | 1.114 | 0.041 | 0.679 |
| | #10 | 117 | 120 | 30 | 57 | 3 | 159 | 1.205 | 1.102 | 1.871 | 1.316 | 0.135 | 1.499 |
| D | #7 | 3 | 27 | 27 | 36 | 3 | 294 | 0.021 | 0.381 | 1.946 | 0.372 | 0.017 | 0.525 |
| | #11 | 3 | 81 | 18 | 51 | 6 | 147 | 2.285 | 1.423 | 2.717 | 0.956 | 0.439 | 1.084 |
| E | #12 | 96 | 90 | 24 | 42 | 12 | 123 | 1.060 | 0.797 | 1.146 | 0.973 | 0.711 | 0.839 |

Note: because the depth of 0.1m at position C is not discussed, the data is "—" in the table.



The above results show that when the IDD is 1.20 g cm$^{-3}$, the response time of PWP is shorter than that of 1.40 g cm$^{-3}$, but the variation of PWP is larger than that of 1.40 g cm$^{-3}$. This shows that as the IDD increases, the response time of PWP will be delayed, and the change of PWP will decrease. The reason is that even if the rainfall intensity is the same, when the slope has diverse hydrological characteristics (such as permeability), the PWP response to rainfall is still significantly different (Lan et al., 2003). For example, the permeability of a slope with a large density is relatively weak, thus, PWP changes are restricted. In addition, as the rainfall intensity increases, the response time of PWP advances, and the variation of PWP increases. The reason is that infiltrated rainwater increases significantly in the early stage of rainfall.

## 3.4 Failure mode of landslides

In the six sets of tests, landslide can be triggered by rainfall. There are similarities in the patterns of slope failures. At the beginning of the rainfall, all rainwater can seep into the slope. There is no surface runoff on the slope, and the soil at the foot of the slope is first softened and slips off. Another similar pattern is that continuous rainfall can cause soil crusts and runoff on the slope surface. Short-term low-lying areas and interlocking ditches will appear due to surface runoff and rainwater erosion. In addition, rainwater infiltration not only causes a water pressure difference between the upper and lower layers of the slope (Zhou et al., 2014), which leads to the first failure of the slope foot, but also causes the migration of fine particles due to subsurface flow which induces the local microstructure of the soil to change.

One of the differences in six tests is the pattern of landslide initiation. In test 1, when the rainfall lasts for 50 minutes, the slope toe is partially unstable, and the soil at the trailing edge continues to slide; when the rainfall lasts for 1402 minutes, all the surface soil slips. The frequent sliding soil is in the shape of a block in test 1. In test 2, when the rainfall lasts for 67 minutes, a small area of soil at the toe of the slope slips; the sliding area slowly spreads to the surroundings; when the rainfall lasts for 717 minutes, the slope has no obvious changes; at the end of rain, the partial right shoulder fails to slide. In test 3, when the rainfall lasts for 32 minutes, a crack with a length of 0.5 m develops on the right slope toe; the soil around the crack slides quickly; when the rainfall lasts for 923 minutes, all the soil on the slope surface is destroyed. In test 4, a low-lying area appears at the foot of the slope when the rainfall lasts for 45 minutes; the surrounding soil gradually slips, and the unstable zone extends to the middle of the slope; since the rainfall lasts for 504 minutes to the end of the rainfall, the slope has stabilized. In test 5, when the rainfall lasts for 26 minutes, the soil on the right slope foot slips; the low-lying areas are enlarged with the continuous rainfall; when the rainfall lasts for 986 minutes, all the soil at the slope toe slips suddenly. In test 6, when the rainfall lasts for 5 minutes, cracks appear on the left foot of the slope, and then the surrounding soil begins to slide rapidly, and the unstable range gradually expands, which induces the repetitive failure of soil. Since the rainfall lasts for 472 minutes to the end of the rainfall, the soil on the right slope shoulder remains stable. The above landslide phenomena show that the landslide initiation has two important characteristics. When the IDD is 1.20 g cm$^{-3}$, the development of cracks leads to soil instability and induces landslide; the formation of landslides is relatively sudden and large in scale. When the IDD is 1.40 g cm$^{-3}$, the soil at the foot of the slope slides first, and then drags the soil at the trailing edge; the failure mode of the landslide is mainly small-scale soil sliding; when the rainfall ends, the soil on the slope has not been completely destroyed.

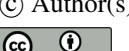



Another difference between six tests is the initiation time of landslide. Table 4 shows that when the slope angle is 45º, and the rainfall intensity is 60 mm h$^{-1}$ along with 90 mm h$^{-1}$, if the IDD increases

from 1.20 g cm$^{-3}$ to 1.40 g cm$^{-3}$, the initiation time of landslide will be delayed by 17 minutes and 13 minutes. When the IDD is 1.20 g cm$^{-3}$, the rainfall intensity is 60 mm h$^{-1}$ and 90 mm h$^{-1}$, if the slope angle increases from 40º to 45º, the starting time will be delayed by 24 minutes and 27 minutes. When the slope is 40º and the IDD is 1.20 g cm$^{-3}$, the starting time of landslide with a rainfall intensity of 90 mm h$^{-1}$ is 21 minutes earlier than that with a rainfall intensity of 60 mm h$^{-1}$. When the slope angle is

45º, the IDD is 1.20 g cm$^{-3}$ and 1.40 g cm$^{-3}$, if the rainfall intensity increases from 60 mm h$^{-1}$ to 90 mm h$^{-1}$, the initiation time will be shortened by 18 minutes and 22 minutes.

Table 4 Initiation time of landslides of six tests.

| Test number | 1 | 2 | 3 | 4 | 5 | 6 |
|---|---|---|---|---|---|---|
| Initiation time (min) | 50 | 67 | 32 | 45 | 26 | 5 |

The above results show that the IDD, slope angle and rainfall intensity have significant effects on

the formation of residual soil landslide. As the IDD increases, the initiation time of the landslide is delayed, the sliding scale of soil decrease, and the landslide initiation type changes from a sudden sliding type to a progressive failure type. The main reason is the energy required to destroy the small pore structure is much greater than the energy required to destroy the large pore structure. Steep slope are not conducive to infiltration of rainwater (Xu et al., 2018), so the initiation time is delayed as the

slope angle increases. In addition, the increase of rainfall intensity can reduce the starting time of landslide, obviously, this is due to the significant increase in infiltrated rainwater at the beginning of the rainfall.

## 4    Discussion

The failure process of the granite residual soil landslide can be classified into five stages. Five

stages are as follows and shown in Table 5.

(i) Rainwater infiltrates. At the beginning of the rainfall, all rainwater penetrates into the soil; at this time, there is no surface runoff on the slope; then, soil moisture and pore water pressure begin to increase; muddy water flows out from the slope toe, and cracks appear at the foot of the slope.

(ii) Soil slides at the slope toe. The interflow converges at the foot of slope, which causes the

shear strength of residual soil to decrease. Consequently, the soil at the slope foot takes the lead in softening and sliding.

(iii) Surface runoff and erosion occur. The water content increases with the continuous rainfall, but the infiltration capacity of the soil decreases. When the shallow soil is saturated, surface runoff gradually forms. Low-lying areas and ditches are developed due to the erosion of surface runoff and

splash erosion of rainfall.

(iv) A steep free face begins to form. The soil at the toe of the slope continues to slide, and the failure range expands to the surroundings, which results in the formation of a steep free surface. Although the soil on the top of the slope has not slipped, it is in an under-stable state.

(v) The soil on the slope top begins to slide. The increase in pore water pressure leads to a decrease

in effective stress and shear strength. In addition, the increase in soil moisture leads to an increase in the sliding force. The soil on the top of the slope begins to slide due to the combined action of the above two factors.



Table 5 Sketch maps and photos of landslide failure process.

| Steps | Rainwater infiltrates | Soil slides at the slope toe | Surface runoff and erosion occur | A steep free face begins to form | The soil on the slope top begins to slide |
|---|---|---|---|---|---|
| Sketch diagram | rainfall infiltration | slipped soil | surface runoff    soil erosion | steep free face | slipped soil |
| Sketch photos | | | | | |


Section 3.2 has pointed out that the changes of volume moisture content mainly include three similar stages: initial constant, significant rise, and stability. However, the response time of pore water pressure at the same position and depth is not synchronized with the water content. During the initial constant and significant rise of water content, the pore water pressure has almost no change or only slight fluctuations. Moreover, pore water pressure begins to increase significantly before moisture content remains stable. Obviously, the time when the pore water pressure begins to increase is behind the time when the moisture content begins to increase. In addition, the change time of pore pressure at a depth of 50 cm is earlier than that at a depth of 30 cm. The main reason is that rainwater infiltration leads to an increase in the weight of the overlying soil, which produces a squeezing effect on the underlying soil, and results in a decrease in void ratio and an increase in pore water pressure.

Section 3.3 has found that the pore water pressure of shallow soil fluctuates significantly during the landslide initiation. It may be related to the difference in the mechanical state of the residual soil. Therefore, the typical periods of the test 2 and the test 3 are selected for discussion. In test 2 with the initial dry density of 1.40 g cm$^{-3}$, when the rainfall lasts for 195-225 minutes, the soil in the slope middle is unstable, which promotes the development of cracks and causes massive soil on the slope to slide (Figure 22a). The seventh sensor is the closest to unstable soil, thus, the data of this sensor is selected for detailed analysis. Figure 22b shows the water content has remained stable during this period, and the soil is in an over-saturated state; however, the pore water pressure gradually increases to a peak when the rainfall duration is 195-201 minutes. Subsequently, pore water pressure decreases rapidly, and maintains a certain degree of volatility. When the rainfall duration is 210 minutes, the pore water pressure begins to increase again. In test 3 with the initial dry density of 1.20 g cm$^{-3}$, when the rainfall lasts for 30-48 minutes, the shallow soil is softened and slides many times (Figure 23a). Figure 23b shows that when the rainfall duration is 30-36 minutes, water content and pore water pressure increase obviously; when the rainfall lasts for 36 minutes, the increasing trend of them is relatively gentle; when the rainfall lasts for 42 minutes, although the pore water pressure increases rapidly again, but the water content remains stable.



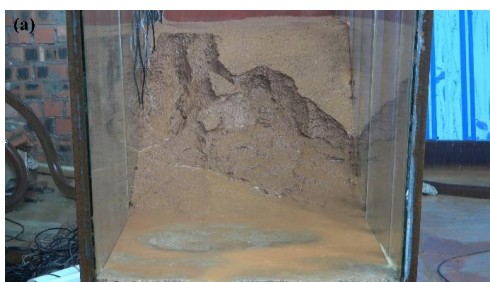
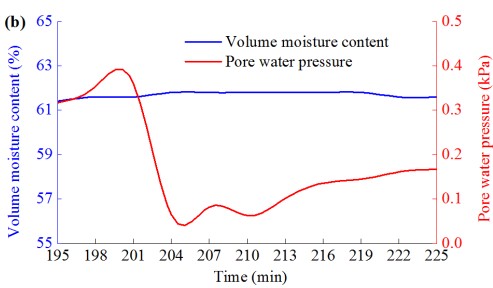

Figure 22 Test phenomenon and data when IDD is 1.40 g cm⁻³. (a) Slope failure phenomenon. (b) Results for sensor #7.

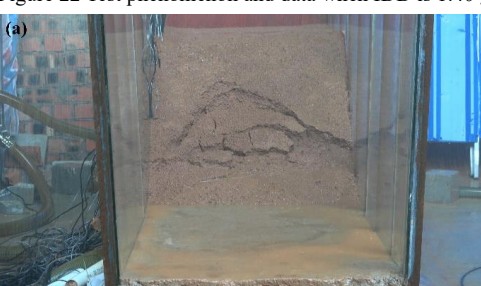
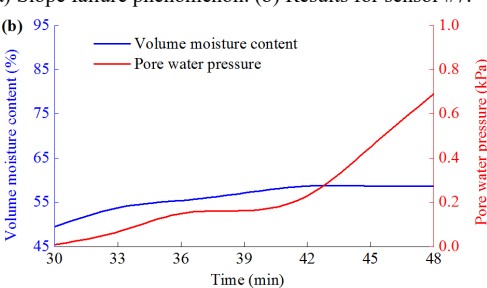


Figure 23 Test phenomenon and data when IDD is 1.20 g cm⁻³. (a) Slope failure phenomenon. (b) Results for sensor #7.

The test results can confirm that the failure process of the landslide is the result of the combined effect of the mechanical properties of shear zone and the pore water pressure (Iverson, 2005). The
property and deformation of the potential shear zone are related to the initial dry density (Iverson et al., 2000), and affect the fluctuation of pore water pressure. When the density is low, the "hammer" effect of rain will compress the shallow soil. At the same time, the deformation of the shear zone is mainly contraction, which causes the void ratio to decrease rapidly and excessive pore water pressure to generate. However, the excess pore water pressure is difficult to dissipate completely in a short time,
which can promote the continuous increase of pore water pressure and the connection of potential sliding surfaces. Therefore, the type of landslide failure is a sudden sliding type in the macroscopic phenomenon (Dai et al., 1999a; Dai et al., 1999b; Mckenna et al., 2011). The increase of initial dry density cannot only inhibit the infiltration rate of rainwater, but also prolong the response time of water content and pore water pressure, and even limit the fluctuation of pore water pressure. As a result, the
ability of the slope to resist seepage damage is improved effectively. Meanwhile, the dilatation of the shear zone causes the pore water pressure to dissipate, which results in the delay of the landslide formation and the recovery of the shear strength. After that, long-term rainfall can restore the loss of pore pressure due to soil dilation, and shear deformation will reappear. At this time, the macroscopic phenomenon of landslide start is progressive (Dai et al., 1999a; Dai et al., 1999b; Mckenna et al., 2011).

**5 Conclusion**

The macroscopic phenomenon of granite residual soil landslide has obvious similarities. At the beginning of the rainfall, all rainwater can seep into the slope. There is no surface runoff on the slope, and the soil at the foot of the slope is first softened and slips off. Another similar pattern is that continuous rainfall can cause soil crusts and runoff on the slope surface. Short-term low-lying areas
and interlocking ditches will appear due to surface runoff and rainwater erosion.



The increase of initial dry density cannot only inhibit the infiltration rate of rainwater, but also prolong the response time of water content and pore water pressure, and even limit the fluctuation of pore water pressure. However, large rainfall intensity will result in a relatively short response time for water content and pore water pressure. In addition, the fluctuation characteristics of pore water pressure may be related to the type of soil shear deformation.

The difference in the starting time and pattern of landslides is related to the initial dry density, slope angle and rainfall intensity. The starting time of a landslide will be delayed as the density and slope angle increase; however, it will be shortened due to the increase in rainfall intensity. When the IDD is 1.20 g cm$^{-3}$, the development of cracks leads to soil instability and induces landslide; the

formation of landslides is relatively sudden and large in scale. When the IDD is 1.40 g cm$^{-3}$, the soil at the foot of the slope slides first, and then drags the soil at the trailing edge; the failure mode of the landslide is mainly small-scale soil sliding; when the rainfall ends, the soil on the slope has not been destroyed completely. This indicates that the failure mode can change from a sudden sliding type to a progressive failure type due to the increase of initial dry density.

In addition, the failure process of the granite residual soil landslide can be classified into five stages: rainwater infiltration, soil sliding at the slope toe, the occurrence of surface runoff and erosion, the formation of a steep free face, and the upper soil sliding. In this process, the response time of pore water pressure at the same position and depth is not synchronized with the water content. Meanwhile, the response time of pore pressure at a depth of 50 cm is earlier than that at a depth of 30 cm. The

above results can confirm the failure process of the landslide is the result of the combined effect of the mechanical properties of shear zone and the pore water pressure.

This paper discusses the failure mode of rainfall-induced landslide of granite residual soil in Southeast Guangxi, which will prepare for the subsequent in-depth analysis of the relationship between soil moisture content and pore water pressure during the failure process of the landslide. In addition,

the dry density of the landslide model in this paper only considers the conditions of shallow soil. Thus, in the future, the author will build a model with different dry densities, and explore the influence of the initial dry density that changes along the depth direction on landslide failure.

**Competing interests**

The authors declare that they have no conflict of interest.

**Acknowledgements**

This research was funded by the National Natural Science Foundation of China (No. 41901132, 51609041), the Natural Scientific Project of Guangxi Zhuang Autonomous Region (No. 2018GXNSFAA138187).

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
