# Peer review of "Failure mode of rainfall-induced landslide of granite residual soil, southeastern Guangxi province, China"

_Earth Surface Dynamics, 2021_

## Author Response (AR1)

**Response to Editors and Referees**

Dear Editors and Referees,

We are very grateful for your help with the manuscript entitled "Failure mode of rainfall-induced landslide of granite residual soil, southeastern Guangxi province, China". Your valuable comments effectively improve the quality of the paper. We have carefully revised the manuscript following your detailed comments and proofread the content to remove mistakes about grammar and spelling in real earnest.

Please find the following response to the comments of the reviewers.

Best wishes,
Shanbai Wu and other authors

**Notes**

1. Comments from the reviewers are listed in **black**.
2. Explanations to the comments are in **blue**.
3. Revisions in the revised manuscript are in **red**.

**Referee 1**

**General comment**

This paper presented a study on the failure mode of rainfall-induced landslide of granite residual soil using artificial flume model tests. The authors observed the macroscopic phenomena of landslides and analyzed moisture content and pore-water pressure. They finally discussed the initiation patterns of landslides. This topic is very interesting and is of great significance for the prevention and early warning of residual soil landslides in southeast Guangxi. The testing program is reasonably designed, and testing results are reliable. However, some aspects should be modified to achieve more agreeable paper.

**Specific comments**

**1.** In section 2.2, the authors selected two dry densities (1.2 g cm$^{-3}$ and 1.4 g cm$^{-3}$). Why two dry densities (1.2 g cm$^{-3}$ and 1.4 g cm$^{-3}$) are selected? Why does the test not consider 1.3 g cm$^{-3}$? Please explain it.
**Authors' response:** Thanks a lot for your comment. The detailed field investigation shows that the granite residual soil landslide in the study area is mainly shallow, and the dry density of the soil is 1.2 ~ 1.4 g cm$^{-3}$ (Wen, 2015). If 1.3 g cm$^{-3}$ is considered in this paper, there is a difference of 0.1 g cm$^{-3}$ between the initial dry densities. At this time, the difference in the mass of each layer of soil will be 5 kg, and the difference in the total mass of the model slope

will be less than 30 kg. This small difference in soil mass can cause the test results to be similar, and may even be unfavorable for the author to identify the influence of initial dry density on the landslide formation. Therefore, in order to highlight the differences in test results, only two dry densities (1.2 g cm$^{-3}$ and 1.4 g cm$^{-3}$) are selected in this paper. The explanation has been added in this paper, it can be found in Line 132-135.

**2.** The size of Figure 5(b) should be normalized. The unit of model size and soil depth should be consistent in the paper.

**Authors' response:** Thank you for your comment. The unit of model size and soil depth in Figure 5(b) has been modified to meters. The revised details can be found in Line 171-172. Furthermore, the unit of model size and soil depth throughout the paper has been unified as meters.

**3.** There are many figures on volumetric water content and pore-water pressure in Sections 3.2 and 3.3. Can they be streamlined according to the content of the paper? For example, the similar change trend can be presented with a figure or a paragraph. In addition, abnormal data in the figures should be clarified.

**Authors' response:** Thanks a lot for your kind suggestion. A-E inside the flume model represents the crest, shoulder, middle, and foot of the slope respectively. The variation characteristics of the volume moisture content and pore water pressure at A, B, and C are relatively similar. Therefore, the figure of C is retained in the revised manuscript to indicate a general trend, while the figure of B is deleted. In addition, the three positions (C, D, and E) are close to the sliding surface. Thus, the data of these three positions are analyzed in Section 3.2, 3.3 and shown in Figure 6-Figure 18. Meanwhile, the interpretation of abnormal data in the figures has been supplemented. The Sections 3.2 and 3.3 have been streamlined and improved. The revised details can be found in Line 237-377.

**4.** In section 3.2, Figure 10(b)-Figure 12(b) show that the fluctuation of volumetric water content occurs in the second rainfall. Please offer a brief explanation for this phenomenon.

**Authors' response:** Thank you for your kind suggestion. At the beginning of the second rainfall in test 4, the maintenance of water pipe caused a short water stop. Therefore, the data fluctuated.

**5.** In Section 3.5, the authors explore the close relationship between the initial dry density, slope angle, rainfall intensity, and landslide initiation. Additionally, the initiation time of six landsides are compared. However, the detailed mechanism is still not clear. A deep discussion is needed.

**Authors' response:** Thanks a lot for your comment. In the discussion section, the mechanism of landslide initiation combined with the initial dry density, slope angle, rainfall intensity, and physical parameters of residual soil has been deeply analyzed. The revised details can be found in Line 420-440.

**6.** In section 4, the authors stated that the failure process of the granite residual soil landslide can be classified into five stages. Is it based on the six tests? Are five stages only summarized

by the six tests?

**Authors' response:** Thank you for your comment. Six model tests have commonness in the patterns of slope failure based on the macroscopic phenomena. Based on these tests, the landslide formation can be classified into five stages and shown in Table 2. They are basically consistent with the results of the field survey in Southeast Guangxi (Wei et al., 2017). Therefore, the failure processes of granite residual soil slopes can be reproduced by flume model tests. The description can be found in Line 379-383.

**7.** The conclusion is not clear enough, such as "the starting time of a landslide will be delayed". Can the quantitative result be offered? In addition, I suggest that some conclusions on the initial conditions can be condensed.

**Authors' response:** Thanks a lot for your kind suggestion. Although the quantitative results for initiation time of landslide have been shown in Table 3, they are supplemented in the conclusion. The content of the conclusion has been streamlined. The revised details can be found in Line 515-517.

**References**

Wei, C., Wen, H., Liao, L., Yang, Y., Ma, S., Zhao, Y., and Chen, L.: Failure characteristics and prevention measures of granite residual soil slope in the southeast of Guangxi Province, China, Earth and Environment, 45, 576-586, 2017 (in Chinese).

Wen, H.: A detailed survey report of geological disasters in Rongxian County, Guangxi., Guangxi Zhuang Autonomous Region Geological Environmental Monitoring Station, Guilin, Guangxi., 196 pp., 2015 (in Chinese).

**Referee 2**

**General comment**

Thank you to allow me to review this manuscript. Granite residual soil landslides are widely distributed in southeastern Guangxi, China. In this paper, the authors reproduced the failure mode and process of granite residual soil landslide by flume model tests. This work is interesting to read. However, some critical points should be addressed before the manuscript can be accepted.

**Specific comments**

(1) The entire manuscript should be polished by a native English speaker, as some sentences are not concise enough and there're some spelling or grammatical mistakes.

**Authors' response:** Thanks a lot for your kind suggestion. The paper has been earnestly streamlined, and the content has been carefully proofread to remove spelling and grammatical errors.

(2) In line 144-145, the particle size distribution of the granite residual soil has an "upward

convex" part in Figure 2, which is impossible in the cumulative distribution curve. The authors should check the accuracy of the data and Figure 2. It is also recommended that the two important particle sizes in the article (2mm and 0.075mm) are marked in Figure 2.

**Authors' response:** Thank you for your comment. We have checked the data of particle gradation, there is no problem with the data. The soil content with a particle diameter between 1mm and 2 mm is relatively small, so the curve has a convex feature. The particle sizes including 2 mm and 0.075 mm have been marked in Figure 2. The revised details can be found in Line 130-131 and 143-144.

(3) In Section "3.1 Macroscopic phenomena of tests", the author intended to elaborate on the macroscopic phenomena of the slope during the 6 experimental tests. However, Figure 6 only shows the slopes after soil sliding in these 6 tests. Therefore, the authors may choose one test as an example, clarify the processes and characteristics of the tests based on the experimental photographs.

**Authors' response:** Thank you for your kind suggestion. Because the macroscopic phenomena in test 3 can fully reflect the whole processes of these six tests, we choose the test 3 to clarify the processes and characteristics of the tests. It can be found in Line 197-205 and 230-236.

(4) In sections "3.2 Moisture Content" and "3.3 Pore Water Pressure", the author explains the variation of the moisture content and pore water pressure during the experiments in detail, but the reasons for the variation have not been analyzed in depth. It is suggested that the author analyze the variations of water content and pore water pressure sensors, combined with the macroscopic phenomenon of the slope failure. The authors should deeply analyze the landslide failure processes, and try to reveal the failure mechanism based on your tests.

**Authors' response:** Thanks a lot for your comment. In sections 3.2 and 3.3 of the revised manuscript, the commonness and differences of the variation of volume moisture content and pore water pressure have been analyzed combined with slope failure. The revised details can be found in Line 237-377. Furthermore, in the discussion section, the failure process of granite residual soil slope has been summarized, and the mechanism of landslide initiation has been comprehensively explored based on the test phenomena, physical parameters of residual soil, and mechanical behavior of the soil. The revised details can be found in Line 379-418.

(5) In line 280-295, in view of the water content value in Test 5 and 6, the authors did not clarify the phenomenon that the water content in Test 6 has a longer response time than Test 5. It can be seen from No. 1~4 tests, the initial response time of the water content will become shorter as the rainfall intensity increasing. However, the water content in Test 6 has a much longer response time than Test 5.

**Authors' response:** Thanks a lot for your kind suggestion. The VMC in test 6 has a longer response time than that in test 5. It is obvious in the slope crest, such as the position C. The worth noting in section 3.1 is that the sliding time of test 6 is earlier than that in test 5. The main reasons of the above abnormal phenomena are including two aspects. One is that when the rainfall intensity is relative larger, much more rainwater can penetrate the soil quickly. This process can cause silt and clay to migrate vertically and accumulate at a certain depth (Fang et al., 2012). Subsequently, the infiltration path will be blocked by the fine particles. Furthermore,

rainwater cannot infiltrate the soil smoothly, and causes the long response time of VMC at the slope crest. The other is that rainfall infiltration can cause a difference in water pressure between the slope crest and the slope foot; this effect of seepage force causes the slope foot to slide first (Zhou et al., 2014). In test 5 and test 6, the soil failures are both found in the slope foot at the beginning of rainfall. It is consistent with the result made by Zhou et al. (2014). This local deformation of the slope can cause internal force unbalance and soil microstructure change. The rainfall infiltration will be affected later (Chang et al., 2021). The revised details can be found in Line 277-294.

(6) The author analyzes the failure modes and processes of the granite residual soil landslide by considering three variables, which are slope angles, initial dry densities and rainfall intensities. Frankly speaking, the experimental tests are too few to support the conclusion of the article. If possible, I suggest the author conduct more tests by considering different test conditions.

**Authors' response:** Thank you for your comment. The test conditions are set based on the results of the field survey (Wen, 2015). Among them, two initial dry densities of 1.20 g cm$^{-3}$ and 1.40 g cm$^{-3}$ are set to highlight the differences in test phenomena. The slope angles in the study area range from 30 ºto 45 º, but the angles of most slopes are 40 ºto 45 º. Thus, the slope angles are set to 40 ºand 45 º. Rainfall intensity, and durations are set based on rainfall data from multiple landslide events in the study area in 2010. The description can be found in Line 132-138. Moreover, the macroscopic phenomena of slope failure are summarized based on six tests. They are basically consistent with the results of the field survey in Southeast Guangxi (Wei et al., 2017) and can be explained by the researches made by Iverson (Iverson, 2005; Iverson et al., 2000) and Dai (Dai et al., 1999a; Dai et al., 1999b; Mckenna et al., 2011). It indicates that the results of flume model tests are reliable. In the future, more model tests involving multiple factors will be conducted through the orthogonal experimental design. The deep relationship between landslide process and the factors will be explored.

(7) The conclusions in this manuscript should be rewritten and rearranged in a more logical way.

**Authors' response:** Thank you for your kind suggestions. The conclusion has been rewritten more logically. The revised details can be found in Line 504-525.

**References**

Chang, Z., Huang, F., Huang, J., Jiang, S., Zhou, C., and Zhu, L.: Experimental study of the failure mode and mechanism of loess fill slopes induced by rainfall, Eng. Geol., 280, 1-16, https://doi.rog/10.1016/j.enggeo.2020.105941, 2021.

Dai, F., Lee, C. F., and Wang, S.: Analysis of rainstorm-induced slide-debris flows on natural terrain of Lantau Island, Hong Kong, Eng. Geol., 51, 279-290, https://doi.org/10.1016/s0013-7952(98)00047-7, 1999a.

Dai, F., Lee, C. F., Wang, S., and Feng, Y.: Stress-strain behaviour of a loosely compacted voleanic-derived soil and its significance to rainfall-induced fill slope failures, Eng. Geol., 53, 359-370, https://doi.org/10.1016/s0013-7952(99)00016-2, 1999b.

Fang, H., Cui, P., Pei, L., and Zhou, X.: Model testing on rainfall-induced landslide of loose soil in Wenchuan earthquake region, Nat. Hazard. Earth. Sys., 12, 527-533, https://doi.org/10.5194/nhess-12-527-2012, 2012.

Iverson, R. M.: Regulation of landslide motion by dilatancy and pore pressure feedback, J. Geophys. Res-Earth., 110, 1-16, https://doi.org/10.1029/2004JF000268, 2005.

Iverson, R. M., Reid, M. E., Iverson, N. R., LaHusen, R. G., and Logan, M.: Acute sensitivity of landslide rates to initial soil porosity, Science, 290, 513-516, https://doi.org/10.1126/science.290.5491.513, 2000.

McKenna, J. P., Santi, P. M., Amblard, X., and Negri, J.: Effects of soil-engineering properties on the failure mode of shallow landslides, Landslides, 9, 215-228, https://doi.org/10.1007/s10346-011-0295-3, 2011.

Wei, C., Wen, H., Liao, L., Yang, Y., Ma, S., Zhao, Y., and Chen, L.: Failure characteristics and prevention measures of granite residual soil slope in the southeast of Guangxi Province, China, Earth and Environment, 45, 576-586, 2017 (in Chinese).

Wen, H.: A detailed survey report of geological disasters in Rongxian County, Guangxi., Guangxi Zhuang Autonomous Region Geological Environmental Monitoring Station, Guilin, Guangxi., 196 pp., 2015 (in Chinese).

Zhou, J., Du, Q., Li, Y., and Zhang, J.: Centrifugal model tests on formation mechanism of landslide-type debris flows of cohesiveless soils, Chinese Journal of Geotechnical Engineering, 36, 2010-2017, 2014 (in Chinese).

---

## Author Response (AR2)

**Response to Editor and Reviewer**

Dear Editor and Reviewer,

We are very grateful for your help with the manuscript entitled "Failure mode of rainfall-induced landslide of granite residual soil, southeastern Guangxi province, China". Your valuable comments effectively improve the quality of the paper. We have carefully revised the manuscript following your detailed comments, as well as proofread the content to remove mistakes about grammar and spelling in real earnest.

Please find the following response to the comments of the reviewer.

Best wishes,
Shanbai Wu and other authors

**Notes**

1. The comments from the reviewer are listed in black.
2. The explanations to the comments are in **blue.**
3. The change made can be found in the marked-up manuscript.

**Reviewer 1**

**General comment**

I do appreciate the efforts the authors devoted to this revision. I almost fully satisfy with this revision. However, I still have some follow up and minor comments as follows.

**Specific comments**

**1.** It can be found that 21 figures are included in this manuscript. Are all of them necessary?
**Authors' response:** Thanks a lot for your comment. After serious thought, we think these 21 figures are necessary. The reasons include four aspects. (1) Figure 1 - 4 show the study area, soil particle gradation, test equipment, and sensor locations, respectively. (2) Figure 5 reflects the failure phenomena of the most representative slope. (3) Since the position C, D, and E is close to the sliding surface, we show the variation of volume moisture content (VMC) and pore water pressure (PWP) at these three positions in curve graphs (Figures 6 to 11, Figures 13 to 18). In addition, for the positions A to E of the six tests in the first rainfall stage, we show the response time of VMC and PWP, stable VMC, and variation of PWP in bar graphs (Figures 12 and 19). (4) The five stages of slope failure are summarized in the discussion section. Based on the mechanical properties of the residual soil, the relationship between the landslide formation and the PWP fluctuation are explored (Figures 20 and 21).

**2.** In Table 2, Figure 5, and 20, it can be found that the slides were tend to occur on the right side of the slope in your experimental tests. Is this related to the locations of sensor were on the right side? If it is, please discuss the limitations of your experimental tests.

**Authors' response:** Thank you for your comment. We have supplemented the relevant discussion in the revised manuscript, which can be found in lines 507 to 519. The detailed content in the revised manuscript is as follows.

Finally, the limitation of the model tests in this paper should be discussed. All sensors are embedded in the center section of the slope (Fig. 4). Therefore, the sensors are less affected by the left or right boundary. Monitoring data are reliable and can reflect the variation of VMC and PWP during landslide formation. Because the sensor is connected to the data collector, the connecting line is embedded in the slope. The surrounding soil is compacted to achieve the preset dry density. However, the influence caused by the material heterogeneity of the connecting line, and the soil cannot be eliminated. The effect is reflected in difference in rainwater infiltration. This may cause the right side of the slope to tend to slide locally (Fig. 5 and Fig. 20). Nevertheless, this trend is temporary and does not dominate the five similar stages of landslide formation. In addition, the five stages are basically consistent with the field survey in Southeast Guangxi (Wei et al., 2017). In conclusion, the model tests in this paper reproduce the failure pattern of granite residual soil slope well. In future research, wireless transmission system will be employed to collect sensor data. This can minimize the disturbance caused by the sensor line.

**References**

Wei, C., Wen, H., Liao, L., Yang, Y., Ma, S., Zhao, Y., and Chen, L.: Failure characteristics and prevention measures of granite residual soil slope in the southeast of Guangxi Province, China, Earth and Environment, 45, 576-586, 2017 (in Chinese).